# Downregulation of miR-182-5p by NFIB promotes NAD+ salvage synthesis in colorectal cancer by targeting NAMPT

Li Zhou [1], Hongtao Liu[1], Zhiji Chen[1], Siyuan Chen[1], Junyu Lu[1], Cao Liu[2], Siqi Liao[1], Song He[1], Shu Chen [3✉] & Zhihang Zhou [1✉]

Nuclear factor I B (NFIB) plays an important role in tumors. Our previous study found that NFIB can promote colorectal cancer (CRC) cell proliferation in acidic environments. However, its biological functions and the underlying mechanism in CRC are incompletely understood. Nicotinamide adenine dinucleotide (NAD+) effectively affects cancer cell proliferation. Nevertheless, the regulatory mechanism of NAD+ synthesis in cancer remains to be elucidated. Here we show NFIB promotes CRC proliferation in vitro and growth in vivo, and downregulation of NFIB can reduce the level of NAD+. In addition, supplementation of NAD+ precursor NMN can recapture cell proliferation in CRC cells with NFIB knockdown. Mechanistically, we identified that NFIB promotes CRC cell proliferation by inhibiting miRNA-182-5p targeting and binding to NAMPT, the NAD+ salvage synthetic rate-limiting enzyme. Our results delineate a combination of high expression of NFIB and NAMPT predicted a clinical poorest prognosis. This work provides potential therapeutic targets for CRC treatment.

[1] Department of Gastroenterology, The Second Affiliated Hospital of Chongqing Medical University, Chongqing 400010, China. [2] Department of Emergency, The General Hospital of Xinjiang Military Command, Urumqi 830000, China. [3] Department of Hematology, The Second Affiliated Hospital of Chongqing Medical University, Chongqing 400010, China. ✉email: chenshu921@163.com; zhouzhihang@cqmu.edu.cn

Colorectal cancer (CRC) is the second most common cancer in women and the third most common in men, with an estimated 1.9 million new cases and 900,000 deaths worldwide in 2020[1]. The incidence of CRC has increased in recent years. Although excellent progress has been made in elucidating the diagnosis, treatment, and molecular mechanisms of CRC and other solid tumors in recent years, the prognosis of CRC patients remains poor. Among newly diagnosed CRC patients, 20% have metastatic disease and another 25% with local disease will develop metastasis later. The 5-year survival rate for patients diagnosed with metastatic CRC is less than 20%[2], and the recurrence rate for patients with advanced stage after aggressive treatment is 28–73%[3]. Therefore, more work should be conducted to gain comprehensive insights into the biology of CRC.

NFIB is a widely studied transcription factor that participates in tumor progression. Recent studies have shown that NFIB promotes proliferation and reduces apoptosis of CRC cells[4], triple-negative breast cancer cells[5], bladder cancer cells[6] and laryngeal squamous cell carcinoma cells[7]. On the other hand, it can inhibit the malignant phenotype of osteosarcoma[8], cutaneous squamous cell carcinoma[9], and lung adenocarcinoma[10]. Our previous research found that NFIB was elevated in acid-adapted CRC cells[11] and promoted tumor cell proliferation, invasion, and drug resistance in sorafenib-resistant hepatocellular carcinoma (HCC) cells[12]. However, we also found that hepatocyte-specific knockout of NFIB facilitated the occurrence of HCC[13]. Therefore, the role of NFIB in the tumor is complex, and its role in CRC needs to be further clarified. Mechanically, NFIB can directly or indirectly regulate gene transcription. Many studies have shown that NFIB can directly promote the transcription of EZH2[14], PINK1[15], RIP2[16], and ERO1A[5] or inhibit the transcription of p21[17], CDK[6], and CDK4[9], and it can also regulate gene expression by regulating chromatin accessibility[18,19]. Recent studies have revealed that NFIB can also regulate noncoding RNAs, such as circ_0082182[20], circ_0026416[21], circMAP7D1[22], lncRNA PVT1[7], and miR-302a[23]. Nonetheless, the mechanism of NFIB in CRC needs to be further explored.

Cancer cell metabolism is geared toward biomass production and unlimited growth[24]. Nicotinamide adenosine dinucleotide (NAD+) has emerged as one of the most important factors involved in both the bioenergetic and regulatory processes[25,26]. NAD+ is an essential electron carrier in redox reactions involved in many metabolic pathways, such as glycolysis, oxidative phosphorylation, the tricarboxylic acid (TCA) cycle, and fatty acid oxidation[27]. NAD+ is also a cofactor of poly (ADP-ribose) polymerase (PARP) and sirtuins, which mediate poly-ADP ribosylation and deacetylation, respectively[28]. The salvage pathway refers to the recycling of nicotinamide (NAM) into nicotinamide mononucleotide (NMN), which can be transformed into NAD+. This pathway is considered to be the main source of intracellular NAD+ because of its high efficiency[29]. Nicotinamide phosphoribosyl-transferase (NAMPT) is the major rate-limiting enzyme for NAD+ salvage biosynthesis[30], which catalyzes NAM to produce NMN. Acting as an essential electron carrier and cofactor, NAD+ has been shown to promote cancer progression[27]. For example, NAMPT increases tumorigenic properties and enriches the cancer-initiating cell population[31], as well as colitis-related colon cancer development[32]. Depletion of NAD+ levels by pharmacological inhibition of NAMPT using FK866 sensitizes several myeloma cell lines to reovirus-induced killing[33]. NAD+-derived interferon γ-induced PD-L1 expression in multiple types of tumors governs tumor immune evasion[34]. The combination of IFN-I and NAMPT inhibition significantly decreased NAD+ levels, pancreatic growth, and liver metastasis in vivo[35]. Our previous work also demonstrated that NAMPT is overexpressed in CRC tissues, and high NAMPT expression levels

are associated with poor prognosis[36]. However, the role of NFIB in NAD+ salvage biosynthesis in colorectal cancer remains to be elucidated.

In the present study, we found that NFIB promotes CRC growth by increasing intracellular NAD+ levels. NFIB post-transcriptionally upregulates the NAD+ salvage synthesis rate-limiting enzyme NAMPT by inhibiting miR-182-5p expression, which could target NAMPT mRNA and inhibit CRC cell proliferation. Finally, both spatial transcriptomic analysis and immunohistochemical staining demonstrated a positive correlation between NFIB and NAMPT expression in CRC specimens, and the subgroup with NFIB[high]/NAMPT[high] expression had the poorest prognosis. In conclusion, the downregulation of miR-182-5p by NFIB promoted NAD+ salvage synthesis by targeting NAMPT in CRC.

## Results

### NFIB promotes CRC cell proliferation, invasion in vitro, and growth in vivo.
We first tested the differential expression levels of NFIB in five CRC cell lines (SW620, HCT116, LoVo, HT29 and SW480) using qRT-PCR, western blotting, and immunofluorescence staining (Supplementary Fig. 1a–c). We then used lentiviral vectors containing shRNA sequences targeting NFIB to knock it down in CRC cell lines with high NFIB expression, i.e. HT29 and SW480 cells (Supplementary Fig. 2a, b); similarly, we overexpressed NFIB in CRC cell lines with low NFIB expression, i.e. SW620 and HCT116 cells (Supplementary Fig. 2c, d). We found that NFIB knockdown significantly inhibited both SW480 and HT29 cell proliferation (Fig.1a, b). NFIB overexpression promoted the proliferation of SW620 and HCT116 cells (Fig. 1c, d). Colony formation was also suppressed following NFIB knockdown (Fig. 1e, f). NFIB depletion decreased the number of invading SW480 cells (Fig. 1g), whereas NFIB overexpression promoted HCT116 cell invasion (Fig. 1h). We further revealed that NFIB knockdown suppressed the in vivo tumor growth of both SW480 (Fig. 1i) and HT29 cells (Fig. 1j). H&E staining showed a larger necrotic region (light pink area) in the control cell-originated tumor tissue than in the knockdown group, reflecting the unlimited growth of cancer cells (Supplementary Fig. 2e). IHC staining revealed fewer Ki67-positive proliferating cells in the NFIB-knockdown group (Supplementary Fig. 2f). Taken together, we found that NFIB promoted CRC cell proliferation, invasion in vitro, and growth in vivo.

### NFIB promotes CRC cell proliferation by influencing the production of NAD+.
Metabolic analysis (LC-MS) was performed to determine the effect of NFIB on CRC metabolism. We found that NAD+ and the related L-tryptophan, nicotinamide, were decreased in NFIB-knockdown HT29 cells (Fig. 2a). These results indicate that NFIB plays an important role in the metabolism of NAD+, and knockdown NFIB can also cause changes in the synthetic pathway (L-tryptophan, De novo synthetic material) and intermediates of NAD (nicotinamide). Subsequent NAD/NADH detection confirmed that intracellular NAD+ was decreased after silencing NFIB in HT29 (Fig. 2b) and SW480 (Fig. 2c) cells. One precursor of NAD+, NMN, was then added to the control and NFIB-knockdown cells to determine whether NAD+ metabolism mediates the function of NFIB. The results showed that NMN supplementation rescued the proliferation of NFIB-knockdown HT29 (Fig. 2d) and SW480 (Fig. 2e) cells. We also confirmed that NMN supplementation remarkably increased intracellular NAD+ levels in both control and NFIB-knockdown cells, although NAD+ was still higher in control cells after NMN treatment (Fig. 2f, g). These results suggested that NFIB facilitates the transformation of NMN to NAD+. In summary, NFIB

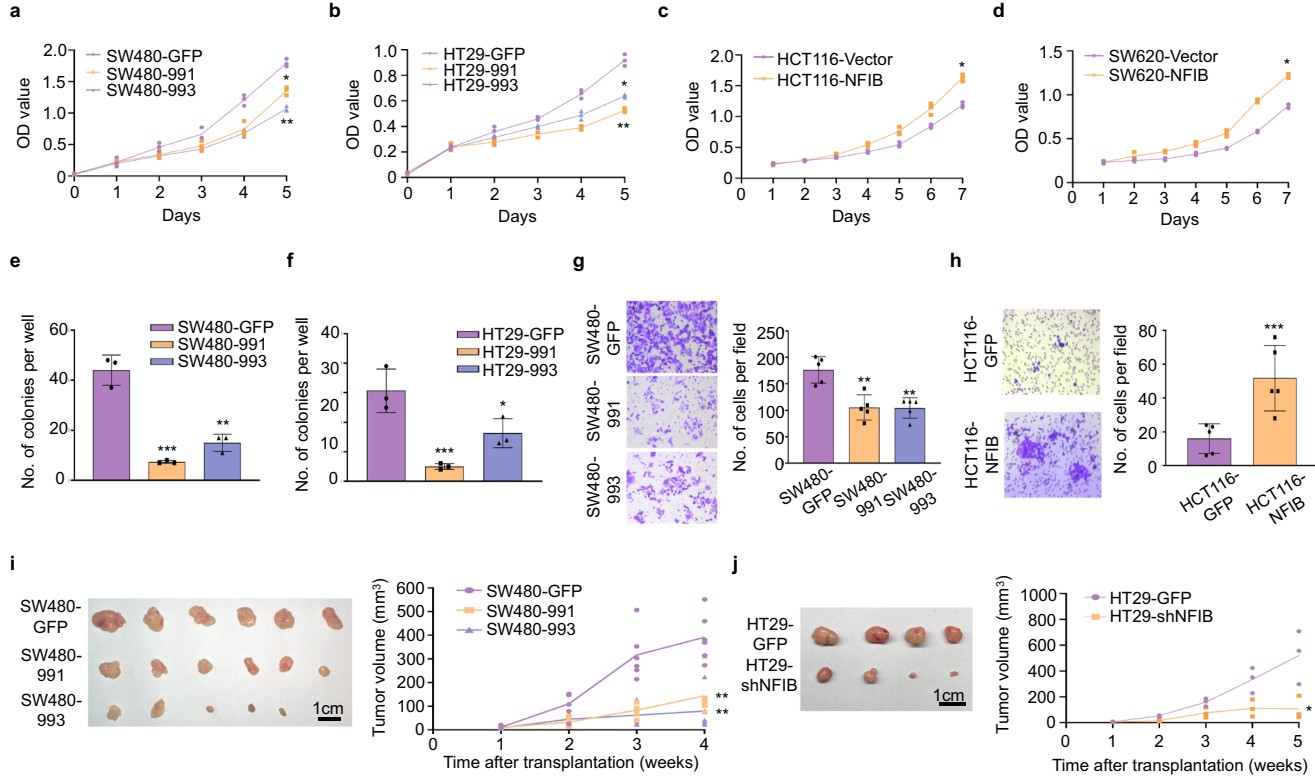

**Fig. 1 NFIB promotes CRC cell proliferation, migration, and invasion in vitro, and growth in vivo. a–d** CCK-8 assays show that knockdown of NFIB inhibited the proliferation of SW480 and HT29 cells, whereas overexpression of NFIB enhanced the proliferation capacity of SW620 and HCT116 cells. **e**, **f** Colony formation experiments showed that NFIB knockdown inhibited the proliferation of SW480 and HT29 cells. **g**, **h** Transwell assays indicated that knockdown of NFIB inhibited the migratory and invasive ability of SW480 cells, while overexpression of NFIB promoted the migratory and invasive capacity of HCT116. **i**, **j** The size of xenograft tumors in the NFIB-knockdown group was significantly smaller than in the control group. The growth curve of xenograft tumors in the NFIB-knockdown group was slower than in the control group. Bar graph data are presented as the mean ± SD; *$P < 0.05$; **$P < 0.01$; ***$P < 0.001$.

promotes CRC cells proliferation by influencing the production of NAD+.

**NFIB post-transcriptionally up-regulates NAMPT in CRC.** Whole transcriptome analysis was performed to explore the mechanism by which NFIB promotes NAD+ synthesis. Differentially expressed genes (DEGs) were enriched in metabolic pathways (Fig. 3a). Interestingly, NAMPT, the rate-limiting enzyme for NAD+ salvage biosynthesis, was among the genes significantly downregulated after NFIB knockdown (Supplementary Fig. 3a). We then confirmed this phenomenon in NFIB-knockdown or NFIB-overexpressing CRC cells. We found that NAMPT was decreased in NFIB-knockdown cells at both mRNA (Fig. 3b) and protein levels (Fig. 3c). Consistently, both NAMPT mRNA and protein levels were increased in NFIB-overexpressing cells (Fig. 3d, e). Spatial transcriptomic analysis showed that the spatial distribution of NFIB and NAMPT was closely correlated in two CRC tissues (Supplementary Fig. 3b). Meanwhile, NFIB and NAMPT were positively correlated with high expression in two CRC tissues (linearly dependent coefficient = 0.126, $p < 0.001$; Fig. 3f). Single-cell RNA sequencing (scRNA-seq) showed a positive correlation (R = 0.097, $P = 1.2e–05$; Fig. 3g) between the expressions of NFIB and NAMPT in epithelium-derived tumor cells (Supplementary Fig. 3c) of CRC. The public CRC database also showed a positive correlation between NFIB and NAMPT expression in bulk CRC tissues ($P < 0.0001$; Fig. 3h). And then, we used ChIP-seq to test whether NFIB could directly promote NAMPT transcription. The ChIP peaks were not concentrated near the transcription start site (TSS) of NAMPT

(Fig. 3i), but mostly located in the distal intergenic and intron regions (Fig. 3j). UCSC, PROMO, GeneCards and JASPAR databases showed that NFIB was not among the transcription factors that could bind to NAMPT (Supplementary Table 1). Therefore, we speculate that NFIB may not directly bind to NAMPT promoters, but regulate NAMPT transcription through other pathways, such as post-transcriptional regulation.

**Downregulation of miRNA-182-5p by NFIB mediated the post-transcriptional regulation of NAMPT.** MicroRNAs are widely involved in tumorigenesis and important post-transcriptional modifications. According to our transcriptomic data, a large number of miRNAs were altered after NFIB knockdown (Fig. 4a). The most upregulated miRNAs are shown in Supplementary Fig. 3d. We validated the sequencing data using qRT-PCR in both SW480 and HT29 cells. The results showed that miR-182-5p was the most significantly upregulated miRNA after NFIB knockdown (Fig. 4b). Three target prediction algorithms (TargetScan, miRDB, and PolymiRTS) were used to predict the binding site between miR-182-5p and NAMPT mRNA (Fig. 4c). To further investigate the regulatory relationship between miRNA-182-5p and NAMPT, miRNA-182-5p mimics and inhibitors were constructed. We found that the miR-182-5p mimic significantly increased the expression level of miR-182-5p (Fig. 4d). Meanwhile, the miR-182-5p mimic and inhibitor decreased and elevated NAMPT expression in HT29 and SW480 cells, respectively (Fig. 4e). MiR-182-5p could thus reduce NAMPT protein abundance in these HT29 and SW480 cells (Fig. 4f). Finally, we generated luciferase reporter plasmids that

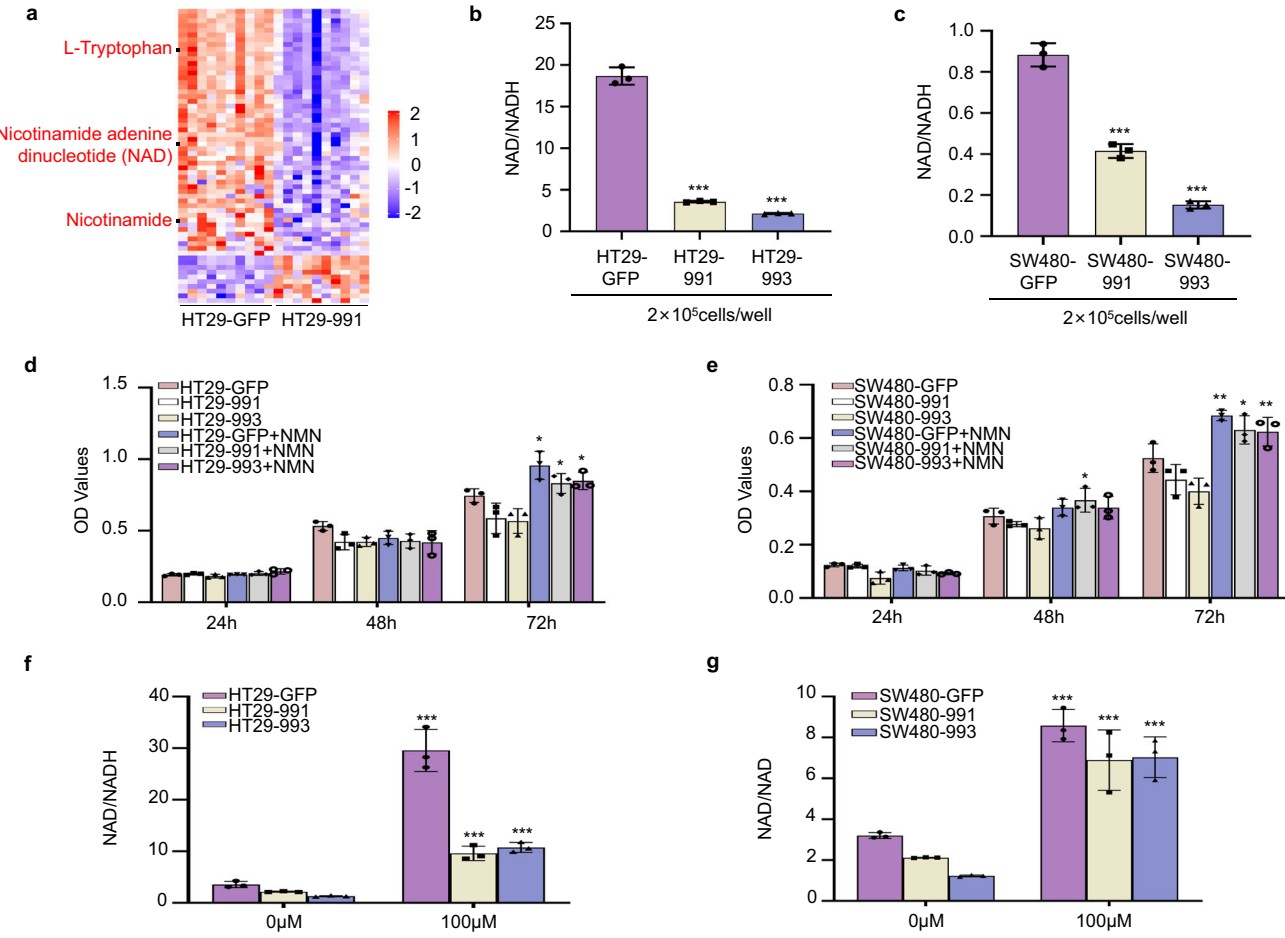

**Fig. 2 NFIB promotes CRC cell proliferation by influencing the production of NAD+. a** LC-MS analysis showed that the content of the NAD metabolic pathway decreased significantly after NFIB knockdown in HT29 cells. **b**, **c** The NAD/NADH level of HT29 and SW480 cells decreased significantly after knocking down NFIB. The CCK-8 assays of HT29 (**d**) or SW480 (**e**) cells were treated with NMN (100 μM) for 3 days. **f**, **g** The NAD+ levels of HT29 or SW480 cells were treated with NMN (100 μM) for 3 days. Bar graph data are presented as the mean ± SD; *$P < 0.05$; **$P < 0.01$; ***$P < 0.001$.

harbor miR-182-5p target sequences in the 3'-UTR (untranslated repeat) of NAMPT or the mutated sequence to diminish binding (Fig. 4g). Using these luciferase reporter plasmids, we found that the miR-182-5p mimics decreased, while its inhibitor increased the activity of the 3′-UTR luciferase reporter of NAMPT in HT29 cells (Fig. 4h). The mutated sequence was not significantly affected by miR-182-5p expression and these effects were also observed in SW480 cells (Fig. 4h). MiR-182-5p inhibited cell proliferation and colony formation in both HT29 (Fig. 5a, b) and SW480 (Fig. 5c, d) cells, which was consistent with a previous study[37]. Finally, we found that the miR-182-5p mimic significantly reduced NAD+ levels, whereas its inhibitor led to higher NAD+ levels in wild-type HT-29 cells (Fig. 5e). Furthermore, the addition of miRNA-182-5p inhibitors in HT29 cells with NFIB knockdown also significantly increased intracellular NAD+ levels (Fig. 5f). Together, these results suggest that the downregulation of miRNA-182-5p by NFIB mediates post-transcriptional regulation of NAMPT to support CRC cell growth.

**High NFIB expression is correlated with advanced TNM stage and poor prognosis of CRC patients**. To explore the role of NFIB in the progression of CRC, we detected NFIB expression in CRC tissues and their corresponding adjacent normal mucosa by IHC ($n = 261$, Fig. 6a). Details of the clinicopathological features of the patients in the study are shown in Table 1. We found that the

expression of NFIB was higher in CRC tissues than in adjacent normal mucosal tissues ($P < 0.001$; Fig. 6b) and increased gradually with TNM staging ($P < 0.05$; Fig. 6c). The Chi-square test revealed that high NFIB expression was significantly correlated with higher T-stage ($P = 0.012$) and lymph node metastasis ($P = 0.028$) in CRC (Supplementary Fig. 4a, b). Furthermore, Kaplan–Meier analysis revealed that CRC patients with higher NFIB expression exhibited reduced overall survival (OS) compared to CRC patients with lower NFIB expression ($P < 0.05$) (Fig. 6d). Meanwhile, CRC patients with higher NFIB expression tended to have shorter disease-free survival (DFS), although the difference was not statistically significant ($P = 0.1063$; Fig. 6e). The subgroup with high expression of both NFIB and NAMPT had the poorest clinical prognoses (Fig. 6f, g). These results indicate that high NFIB expression correlates with CRC progression, and NFIB[high]/NAMPT[high] indicates the poorest prognosis.

**Discussion**

NFIB mainly acts as a "paradox" that has been implicated in the progression of multiple types of tumors[38]. In the present study, we report for the first time that NFIB could promote NAD+ biosynthesis by upregulating the rate-limiting enzyme NAMPT. NAD+ is important for both bioenergetic and regulatory processes[25,26]. The salvage synthesis pathway is considered the main source of intracellular NAD+ because it is more efficient and productive than the de novo synthesis pathway[29]. NAMPT is

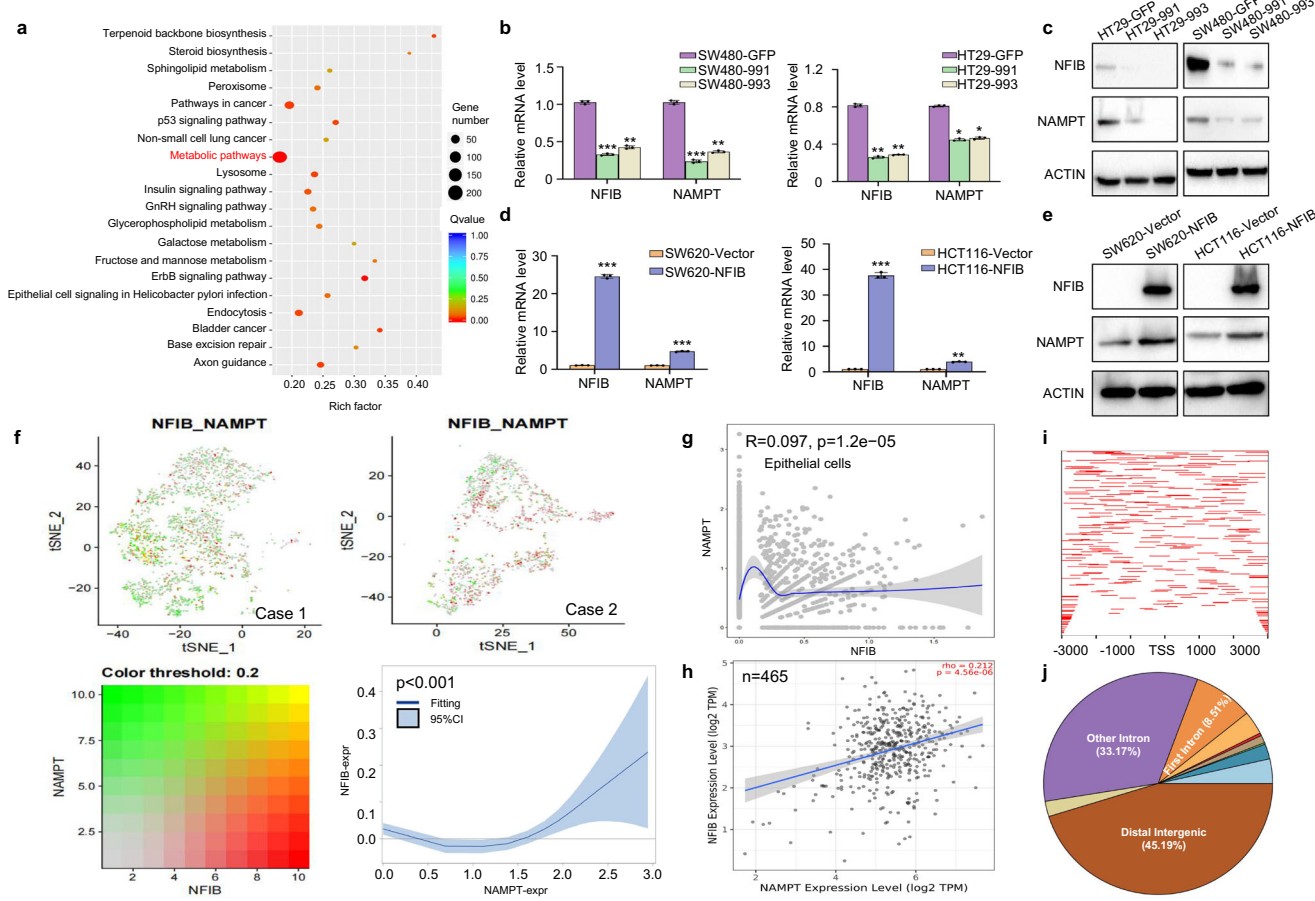

**Fig. 3 NFIB post-transcriptionally up-regulates NAMPT in CRC. a** Sequencing results of the metabolome of knockdown NFIB cells. qRT-PCR and western blot results showed that SW480 and HT29 cells knocked down NFIB while the mRNA (**b**) and protein (**c**) expression level of NAMPT was also decreased. qRT-PCR, and western blot results showed that SW620 and HCT116 cells overexpressed NFIB while the mRNA (**d**) and protein (**e**) expression level of NAMPT was also increased. **f** The t-SNE dimension reduction showed the spatial distribution of NFIB and NAMPT are very close, and the high expression of NFIB and NAMPT was positively correlated (linearly dependent coefficient=0.126, $p < 0.001$). **g** ScRNA-seq analysis of 4 CRC tumor tissues (TNM stage: I, II, III, IV) from GSE132465 showed a positive correlation between the expression of NFIB and NAMPT in epithelial-derived tumor cells ($R = 0.097$, $p = 1.2e-05$). **h** Correlation analysis of NFIB expression and NAMPT expression in COAD patients in the TCGA database (Timer 2.0). **i** Distribution of NFIB binding peak near transcription start site (TSS). **j** The pie chart of NFIB binding peak distribution range proportion. Bar graph data are presented as the mean ± SD; *$P < 0.05$; **$P < 0.01$; ***$P < 0.001$.

the major rate-limiting enzyme in NAD+ salvage biosynthesis, which can promote cancer cell proliferation, migration, stemness, and metastasis[39]. Thus, NAMPT has become a promising anticancer target to deplete NAD+ and impair cellular metabolism. The expression of NAMPT is regulated by the STAT1 transcription factor in macrophages[40], miR-34a in bone marrow mesenchymal stem cells[41], miR-182 in ligamentum flavum cells[42] and miR-149-5p in chondrocytes[43]. The regulatory mechanism of NAMPT in cancer is relatively poorly understood. Transcription factors POU2F2, miR-548b-3p, and miR-206 regulate NAMPT expression in breast cancer cells[44]. Moreover, NAMPT is a direct substrate of SIRT6 deacetylation, increasing NAMPT enzymatic activity in breast cancer https://doi.org/10.1096/fj.201800321R. We are the first time to reveal that NFIB could increase NAMPT expression in CRC cells.

NFIB can directly regulate gene transcription as a transcription factor or indirectly by enhancing chromatin accessibility or post-translational modifications. NFIB can directly promote the transcription of EZH2[14], PINK1[15], RIP2[16], and ERO1A[5], or inhibit the transcription of p21[17], CDK6, and CDK4[9]. Our ChIP-seq results showed that the binding region of NFIB to DNA is not mainly located near TSS, implying that NFIB may not directly

combine with NAMPT in regulating its transcription. Our transcriptomic data showed that NFIB knockdown resulted in significant changes in miRNA expression. Consistent with our findings, recent studies have revealed that NFIB could also regulate non-coding RNAs such as circMAP7D1[22], lncRNA PVT1[7], and miR-302a[23].

MicroRNAs (miRNAs) are a class of evolutionarily conserved small non-coding RNA molecules (approximately 19–23 nucleotides) that are involved in the post-transcriptional regulation of gene expression through imperfect base pairing with the 3'-untranslated region of the target RNAs[45]. Alterations in miRNA expression profiles are common in all human cancers, indicating the importance of such alterations in cancer progression and prognosis[46]. Previous studies have found that miR-182-5p is more highly expressed in normal tissues and intestinal epithelial cells than in tumor tissues and CRC cells[37,47–50], which is consistent with our research. Two studies have also shown that miR-182-5p is elevated in CRC tissues compared to tumor-adjacent tissues. However, they found no significant difference in miR-182-5p expression between colorectal adenocarcinoma and low or medium-differentiated carcinoma[51,52]. MiR-182-5p was selected as the mediator of NFIB-induced

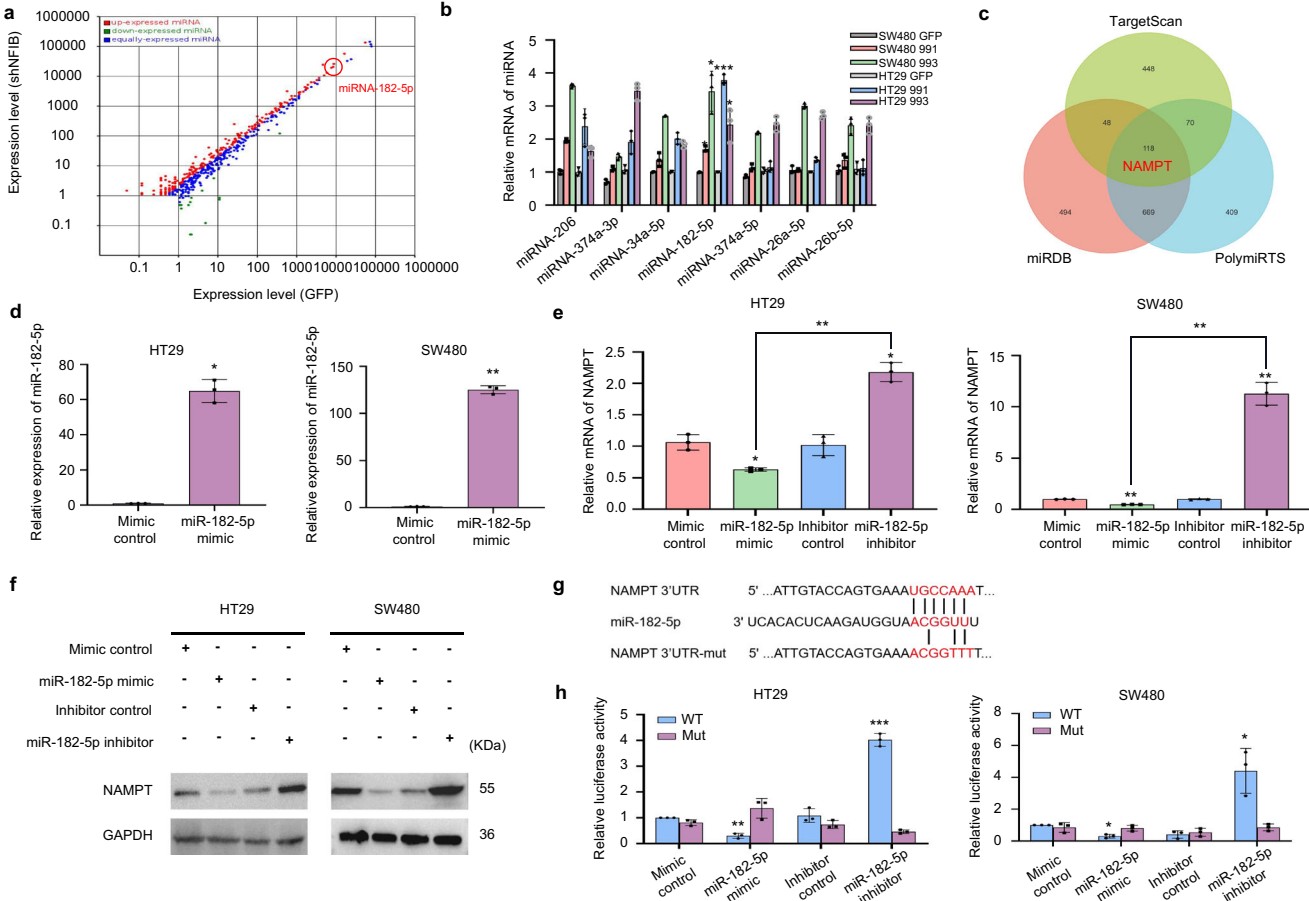

**Fig. 4 Downregulation of miRNA-182-5p by NFIB mediated the post-transcriptional regulation of NAMPT. a** NFIB knockdown cell transcriptome sequencing result. **b** qRT-PCR was used to detect miRNA significantly upregulated after NFIB knockdown. **c** Venn diagram for overlap among the three databases that predict miRNA target genes (TargetScan, miRDB, and PolymiRTs). **d** qRT-PCR for expression of miR-182-5p in HT29 or SW480 cells after transfection of mimic or control. **e** qRT-PCR for expression of NAMPT in HT29 or SW480 cells after transfection of miR-182-5p mimic, mimic control, inhibitor, and inhibitor control. **f** Western blot assays showed that transfection miR-182-5p mimics inhibit protein expression of NAMPT, while transfection miR-182-5p inhibitors promote NAMPT expression in HT29 and SW480 cells. **g** Schematic illustration of NFIB 3'UTR pmirGLO luciferase reporter vectors. **h** Luciferase activity of WT or Mut pmirGLO-NAMPT after transfection with miR-182-5p mimic, mimic control, inhibitor, and inhibitor control in HT29 and SW480 cells. Bar graph data are presented as the mean ± SD; *$P < 0.05$; **$P < 0.01$; ***$P < 0.001$.

NAMPT expression by bioinformatic prediction among the differentially expressed miRNAs after NFIB knockdown. We further validated that miR-182-5p could target the NAMPT mRNA sequence, leading to its downregulation. The function of miR-182-5p depends on the cancer type. MiR-182-5p inhibits the proliferation, migration, and invasion of clear cell renal cell carcinoma[53], bladder tumor[54] and lung adenocarcinoma cells[55]. It also inhibits the proliferation and metastasis of CRC by targeting MTDH36 or suppresses angiogenesis and lymphangiogenesis by directly downregulating VEGF-C[49]. In contrast, miRNA-182-5p promoted the proliferation and migration of bladder cancer[56], glioma[57], and hepatocellular carcinoma[58]. In this study, we identified NAMPT as a novel target of miRNA-182-5p in CRC.

We found that NFIB is highly expressed in CRC, but its regulatory mechanism remains incompletely studied. C-Myc is associated with drug resistance and tumor stem cell subtypes in CRC[59]. In SCLC, c-MYC has been reported to bind directly to the NFIB promoter and influence metastasis[60], while silencing NFIB reduces medulloblastoma stem cell phenotypes[61]. Further research in CRC is needed to corroborate these findings to improve our mechanistic understanding of how NFIB influences aggressiveness in CRC.

In conclusion, NFIB promotes CRC growth by increasing intracellular NAD+ levels, which involves the post-transcriptional upregulation of NAMPT via inhibition of miR-182-5p expression (Fig. 7). Our findings provide novel insights into the regulatory mechanisms underlying NAD+ salvage biosynthesis in CRC.

## Methods

**Cell lines.** Human CRC cell lines (HT29, HCT-116, SW620, SW480, LoVo and LS174T) were purchased from the Culture Collection of the Chinese Academy of Sciences (Shanghai, China). LoVo and SW480 cells were maintained in RPMI-1640 medium (Hyclone, USA) supplemented with 10% fetal bovine serum (Gibco, USA). HT29, HCT-116, SW620, and LS174T cell lines were maintained in DMEM (Gibco, Grand Island, NY, USA) supplemented with 10% FBS (Gibco, USA) and cultured in a 37 °C incubator containing 5% $CO_2$. After 2–3 stable generations, cells were used for subsequent experiments and analyses.

**Lentivirus transfection or plasmid transduction.** Lentiviral particles containing shRNA sequences targeting NFIB (shRNA-991: GCCACATCATATCACAGTA; shRNA-993: CCAATTGGAGAAATCCCAA) or the NFIB coding sequence (NM_005596) were purchased from Obio (China). The cells were seeded at 50–60% confluence and then they were transfected with lentivirus particles. Puromycin (Sigma-Aldrich, St Louis, MO, USA) was used to select stable clones 72 h after transfection. Knockdown efficiency was validated using qRT-PCR and western blotting. MiR-182-5p microRNA mimic, mimic control, inhibitor, and inhibitor control were synthesized by Genepharma (China). HT29 and SW480 cells

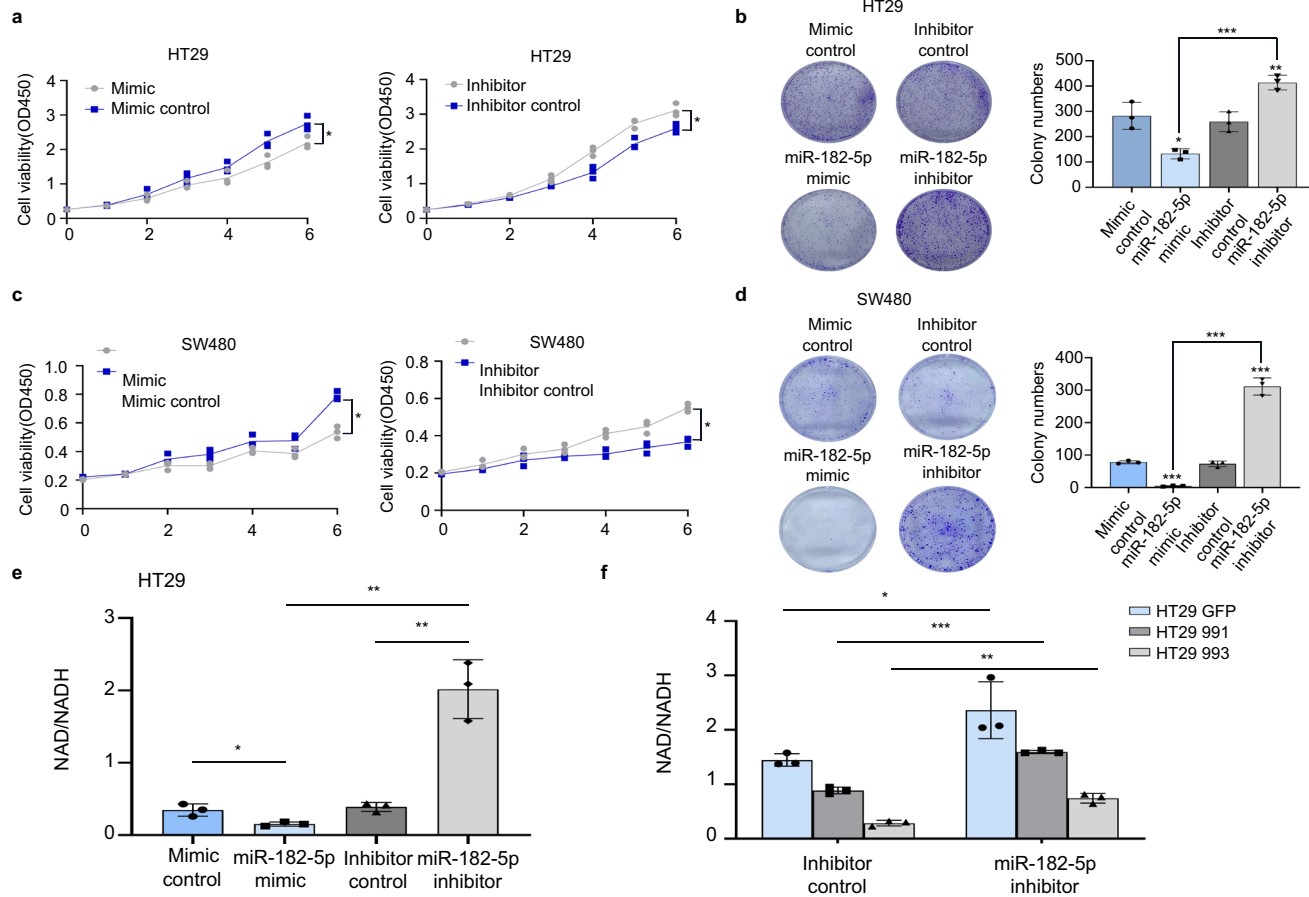

**Fig. 5 MiRNA-182-5p inhibits CRC cell proliferation. a** CCK-8 assay showed HT29 cell proliferation after transfected with miR-182-5p mimic, mimic control, inhibitor, or inhibitor control. **b** Colony formation assay of HT29 cells after transfected with miR-182-5p mimic, mimic control, inhibitor, and inhibitor control. **c** CCK-8 assay showed SW480 cells proliferation after transfection. **d** Colony formation assay of SW480 cells after transfection. **e** MiR-182-5p mimic could significantly reduce NAD+ level and inhibitor of miR-182-5p led to higher NAD+ level in HT29 cells. **f** MiR-182-5p inhibitor could significantly increase NAD+ level in HT29 NFIB knockdown cells. Bar graph data are presented as the mean ± SD; *$P < 0.05$; **$P < 0.01$; ***$P < 0.001$.

were transfected with plasmids using Lipo8000™ Transfection Reagent (Beyotime, China), according to the manufacturer's instructions. After 48 h, transfected cells were harvested for in vitro functional experiments or RNA isolation. After 72 h, the transfected cells were harvested for protein extraction.

**Quantitative RT-PCR.** Total RNA from the cells was prepared using a Trizol reagent according to the manufacturer's protocol (TaKaRa, China). Reverse transcription of total RNA to cDNA was performed using a first-strand cDNA synthesis kit (Roche Diagnostics, USA). Quantitative real-time PCR analysis was performed using SYBR Select Master Mix (ThermoFisher, USA) and gene-specific primers (Genepharma, China) on an ABI 7500 Fast Real-Time PCR system (Applied Biosystems, USA). mRNA expression data were normalized to that of GAPDH. For miRNA detection, total RNA and miRNA were isolated with Trizol reagent using DirectzolTM RNA Miniprep (Zymo Research, CA, USA). RNA was reverse-transcribed using the miScript Reverse Transcription kit (Qiagen, Germany) according to the manufacturer's instructions. Quantitative PCR was performed using the miScript SYBR Green PCR kit (Qiagen, Germany) and miScript Primer assays (Qiagen, Germany) for miRNAs and RNU6 as an internal control. miRNAs were detected on ABI 7500 Fast real-time PCR system (Applied Biosystems, Foster City, CA, USA). Relative expression values were normalized to the internal control using the 2-ΔΔ Ct method. Primer sequences are listed in Supplementary Table 2.

**Protein extraction and western blot.** Cells were lysed with RIPA lysis buffer; cell lysates were then subjected to an 8–12% sodium dodecyl sulfate–polyacrylamide gel electrophoresis (SDS-PAGE), transferred onto polyvinylidene difluoride membrane (Bio-Rad, USA), and probed with antibodies. The antibodies used here were as follows: NFIB (Abcam, UK, 1:1000 dilution), NAMPT (Abcam, UK, 1:1000 dilution), and GAPDH (Proteinates, China).

**Transwell migration and invasion assays.** Cells ($5 \times 10^4$) cells in a 500 μl volume of serum-free medium were placed in the upper chambers of a Transwell (BD, USA) and incubated at 37 °C for 16 h for the migration or invasion assay. The cells that penetrated the uncoated (migration) or Matrigel (BD, USA)-coated (invasion) filters were counted at a magnification of ×200 in 15 randomly selected fields and the mean number of cells per field was recorded.

**Immunofluorescence staining.** HT29, HCT116, SW480 and SW620 cells were seeded onto coverslips at the bottom of the wells. Cells were fixed with 4% paraformaldehyde for 30 min, permeabilized with Triton X-100 for 10 min, and blocked with 2.5% bovine serum albumin for 1 h. Cells were then incubated with the antibody against NFIB antibody (Abcam, UK, 1:100 dilution) overnight at 4 °C, followed by incubation with Alexa Fluor 488-conjugated secondary antibody for 1 h at room temperature. Cell nuclei were stained with DAPI for 5 min. Images were captured using an inverted fluorescence microscope (PerkinElmer, USA).

**Immunohistochemical (IHC) staining.** The tissue microarray was used for IHC staining as previously described[39]. Briefly, paraffin sections of the tissue microarray were dewaxed, deparaffinized in xylene, and rehydrated through a graded ethanol series. After heating in a pressure cooker for 5 min for antigen retrieval, the sections were subjected to endogenous peroxidase elimination. Primary antibodies against NFIB (Abcam, UK) and Ki67 (Proteintech, China) were added to the sections and incubated overnight at 4 °C. The sections were then incubated with horseradish peroxidase (HRP)-conjugated secondary antibodies (ZSGB-BIO, China) for 1 h at 37 °C and stained with diaminobenzidine (BOSTER, China) and hematoxylin. Sections incubated with PBS were used as negative controls. The immunoreactive percentage and intensity were scored. The percentage of positive cells was graded as 0, <5%; +1, 5–25%; +2, 26–50%; +3, 51–75%; and +4, 76–100% positive cells, and the intensity of cellular staining was scored as 0, negative; +1, weak; +2, moderate; and +3, strong. The final staining score was obtained by multiplying the two scores.

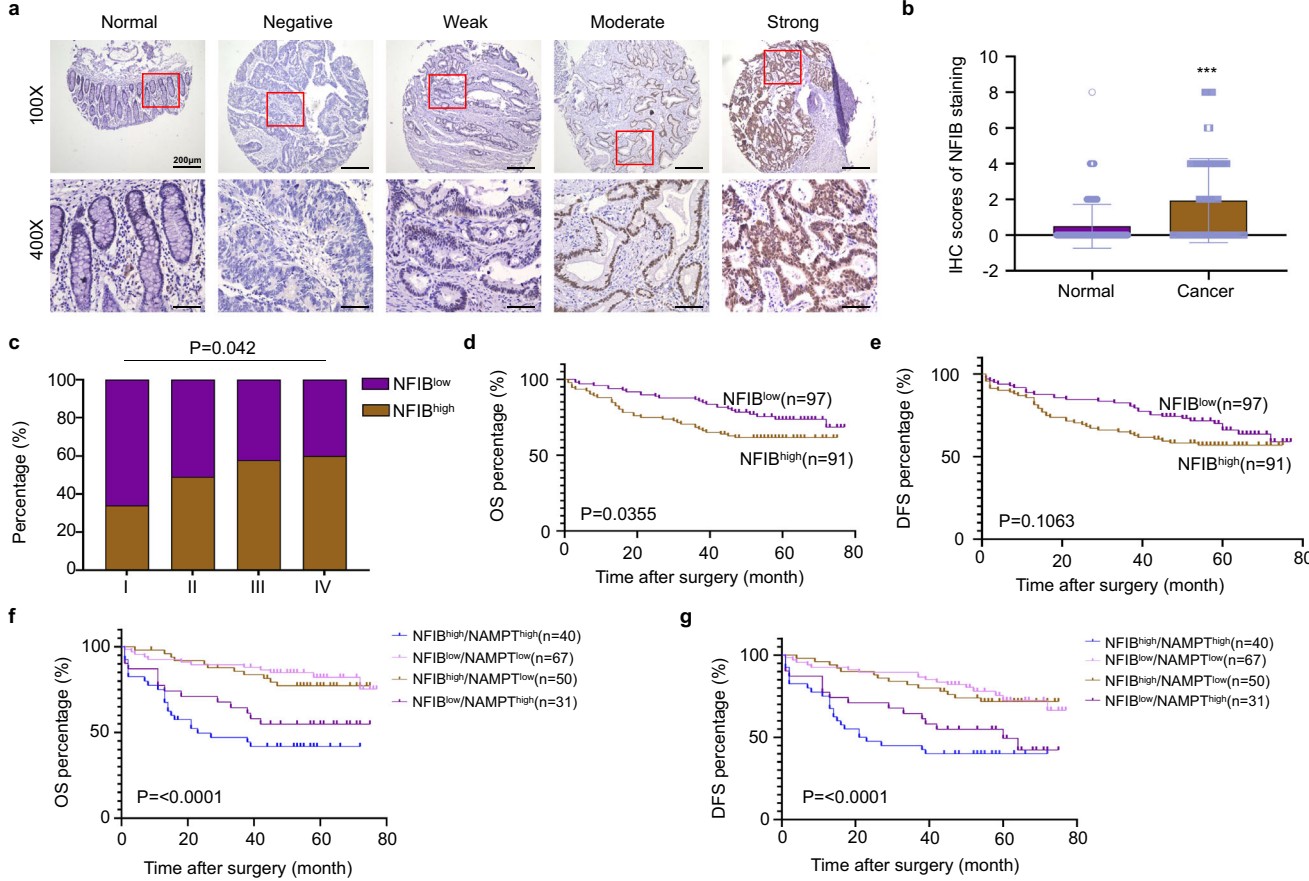

**Fig. 6 High NFIB expression is correlated with advanced TNM stage and poor prognosis of CRC patients. a** The NFIB expression was determined by IHC in CRC tissue and adjacent normal tissues ($n = 261$). Representative IHC images of normal mucosa, negative, weak, moderate, and strong NFIB expression are shown. **b** IHC scores showed overexpression of NFIB in CRC compared with normal colorectal tissues ($p < 0.001$). **c** NFIB expression level increased with the TNM stage ($p = 0.042$). **d, e** OS and DFS curves of patients with low NFIB protein expression ($n = 91$) and those with high NFIB protein expression ($n = 97$). High NFIB expression is associated with poor prognosis ($p = 0.0355$). **f, g** OS and DFS curves of patients with high NFIB and NAMPT protein expression. High NFIB and NAMPT levels are associated with poor prognosis ($n = 188$). Scale bar: 200 μm. IHC immunohistochemical, OS overall survival, DFS disease-free survival. Bar graph data are presented as the mean ± SD; ***$P < 0.001$.

**Whole transcriptomic analysis.** Total RNA from NFIB-knockdown HT29 or control cells was extracted and sequenced by Vazyme Corporation (Nanjing, China) to reveal differentially expressed mRNAs, lncRNAs, circRNAs, and small RNAs. After removing rRNA using the Ribo-Zero kit (Illumina, USA), the cDNA library was constructed using VAHTSTM DNA Clean Beads (Vazyme, China). The final libraries were quantified using an Agilent 2100 Bioanalyzer (Agilent, USA) and an ABI Step One Plus Real-Time PCR system. The libraries were then sequenced on a HiSeq 2500 platform (Illumina, USA). For small RNA sequencing, small RNA was isolated from total RNA using 15% PAGE. The small RNA library was constructed using the TruSeq Small RNA Sample Preparation Kit (Illumina, USA). The libraries were then sequenced on a Hiseq 2500 platform (Illumina, USA) using a synthetic method. Differentially expressed genes (DEGs) were assessed using Cuffdiff software. DEGs between the two groups were considered significant when the absolute log2 signal ratio was larger than 1.0, with a $p$ value < 0.05. DEGs were enriched according to the KEGG database.

**Metabolic analysis.** Metabolic analysis was performed using APTbiotech (Shanghai, China), with 10 replicates for each group (HT29-shNFIB or control). Briefly, $5 \times 10^7$ cells were processed using an ultrasonic homogenizer in a mixture of methanol, acetonitrile, and water (2:2:1, vol/vol). The dried supernatant was subjected to Liquid Chromatography-Mass Spectrometry (LC-MS) analysis. The samples were separated by Agilent 1290 Infinity LC system and detected using a Triple TOF 5600 mass spectrometer. The obtained raw data were transferred to mzXML format via ProteoWizard. The data were then imported into the software XCMS, which performs peak detection, peak identification, and peak alignment. Variable Importance of Projection (VIP) values obtained from the OPLS-DA model was used to rank each variable's overall contribution to group discrimination. Differential metabolites were selected with VIP values greater than 1.0 and p-values less than 0.05.

**Chromatin immunoprecipitation sequencing (ChIP-seq).** The ChIP assay was performed using a ChIP assay kit (Cell Signaling Technology, USA) in SW620 cells. Briefly, cells were crosslinked with formaldehyde, and chromatin was fragmented by digestion with micrococcal nuclease and sonication. Anti-NFIB or rabbit anti-IgG antibodies were used for precipitation, and immunoprecipitates were purified using Agarose Beads. The purified ChIP DNA was adapter-ligated and PCR-amplified according to the manufacturer's instructions (Illumina, San Diego, CA, USA). The final libraries were quantified using an Agilent 2100 Bioanalyzer (Agilent, USA) and KAPA kit (Cat no. KK4602, UK). The libraries were then sequenced on a Hiseq 2500 platform (Illumina, USA). Peaking calling was performed based on the filtered sequence data. The peaks were further annotated according to the reference genome.

**Spatial transcriptomic analysis.** Four fresh CRC tissues were subjected to 10× spatial transcriptomic analysis (Oebiotech, China). OTC-embedded tissues were sectioned and stained with hematoxylin and eosin to optimize the target region. After fixation and permeabilization, mRNA in the cells was released and bound to the corresponding capture probes. cDNA synthesis and library preparation were performed using captured RNA as a template. NGS was performed on sequencing libraries. We used the Loupe Browser software to visualize gene expression.

**Single-cell RNA sequencing (scRNA-seq) and data processing.** The CRC single-cell database was used to obtain information on 4 tumor tissues in GSE132465 (SMC03-T-TNM-IIIC, SMC10-T-TNM-IIA, SMC24-T-TNM-I, and SMC25-T-TNM-IVA). Detailed scRNA-seq and data processing procedures are referred to the literature[62]. Use metadata annotated cell types for gene correlation analysis and this part was conducted by OE biotech Co., Ltd. (Shanghai, China).

**NAD/NADH detection.** NAD/NADH ratio was detected according to the manufacturer's instructions (Abcam, ab65348, UK). NAD/NADH was extracted from

| Features | No. of patients (%) | NFIB level | | P value |
|---|---|---|---|---|
| | | Low (*n* = 131) No. (%) | High (*n* = 130) No. (%) | |
| **Gender** | | | | |
| Male | 153(58.6%) | 83(54.2%) | 70(45.8%) | 0.119 |
| Female | 108(41.4%) | 48(44.4%) | 60(55.6%) | |
| **Age** | | | | |
| ≤60 | 131(50.2%) | 73(55.7%) | 58(44.3%) | 0.073 |
| >60 | 130(49.8%) | 58(44.6%) | 72(55.4%) | |
| **NAMPT level** | | | | |
| Low | 168(64.4%) | 92(54.8%) | 76(45.2%) | 0.047 |
| High | 93(35.6%) | 39(41.9%) | 54(58.1%) | |
| **Vascular invasion** | | | | |
| Yes | 99(37.9%) | 51(51.5%) | 48(48.5%) | 0.738 |
| No | 162(62.1%) | 80(49.4%) | 82(50.6%) | |
| **Nerve invasion** | | | | |
| Yes | 57(21.8%) | 30(52.6%) | 27(47.4%) | 0.677 |
| No | 204(78.2%) | 101(49.5%) | 103(50.5%) | |
| **Differentiation degree** | | | | |
| Well | 69(26.4%) | 38(55.1%) | 31(44.9%) | 0.06 |
| Moderate | 98(37.5%) | 55(56.1%) | 43(43.9%) | |
| Poor | 94(36.0%) | 38(40.4%) | 56(59.6%) | |
| **Size (cm)** | | | | |
| ≤4.8 | 142(54.4%) | 74(52.1%) | 68(47.9%) | 0.498 |
| >4.8 | 119(45.6%) | 57(47.9%) | 62(52.1%) | |
| **T stage** | | | | |
| I | 8(3.1%) | 5(62.5%) | 3(37.5%) | 0.056 |
| II | 49(18.8%) | 32(65.3%) | 17(34.7%) | |
| III | 189(72.4%) | 89(47.1%) | 100(52.9%) | |
| IV | 15(5.7%) | 5(33.3%) | 10(66.7%) | |
| **T stage** | | | | |
| I + II | 57(21.8%) | 37(64.9%) | 20(35.1%) | 0.012 |
| III + IV | 204(78.2%) | 94(46.1%) | 110(53.9%) | |
| **N stage** | | | | |
| 0 | 156(59.8%) | 87(55.8%) | 69(44.2%) | 0.06 |
| 1 | 65(24.9%) | 25(38.5%) | 40(61.5%) | |
| 2 | 40(15.3%) | 19(47.5%) | 21(52.5%) | |
| **N stage** | | | | |
| Negative | 156(59.8%) | 87(55.8%) | 69(44.2%) | 0.028 |
| Positive | 105(40.2%) | 44(41.9%) | 61(58.1%) | |
| **Distant metastasis** | | | | |
| Negative | 256(98.1%) | 129(50.4%) | 127(49.6%) | 0.684 |
| Positive | 5(1.9%) | 2(40.0%) | 3(60.0%) | |
| **TNM stage** | | | | |
| I | 50(19.2%) | 33(66.0%) | 17(34.0%) | 0.042 |
| II | 104(39.8%) | 53(51.0%) | 51(49.0%) | |
| III | 102(39.1%) | 43(42.2%) | 59(57.8%) | |
| IV | 5(1.9%) | 2(40.0%) | 3(60.0%) | |
| **TNM stage** | | | | |
| I + II | 154(59.0%) | 86(55.8%) | 68(44.2%) | 0.028 |
| III + IV | 107(41.0%) | 45(42.1%) | 62(57.9%) | |

**Table. 1 Correlation between NFIB expression and clinicopathological features in cancer tissues from 261 CRC patients.**

$2 \times 10^6$ cells using NAD/NADH Extraction Buffer. The supernatant was collected and subjected to enzyme removal using a 10kD spin column (Abcam, ab93349, UK). NAD was decomposed by heating at 60 °C for 30 min for NADH detection. NAD cycling enzyme mix and NAD cycling buffer were added to the samples to detect NAD and NADH at OD 450 nm.

**Luciferase reporter assay.** HT29 and SW480 cells were seeded in 24-well plates overnight and then co-transfected with pmirGLO-NAMPT 3'-UTR or MUT and miR-182 mimic, mimic control, miR-182 inhibitor, and inhibitor control. After 48-h of culture, firefly and Renilla luciferase activities were measured using the Dual-Lumi™ Luciferase Assay Kit (Beyotime, China), according to the manufacturer's instructions, and firefly luciferase activity was standardized by Renilla luciferase activity.

**Cell proliferation assay.** For the cell proliferation assay, $1 \times 10^3$ cells were plated in 96-well plates in each well with three replicates. At 0, 24, 48, 72,96, and 120 h after transfection, 10 μl of CCK-8 reagent (Dojindo Molecular Technology, Japan) was added to each well with the cell culture medium for 2 h of incubation.

The absorbance was measured at 450 nm using a microplate reader (Eppendorf, Germany).

**Subcutaneous transplantation.** BALB/c athymic male nude mice (4 weeks old) were obtained from Charles River Biotechnology (Beijing, China). NFIB-knockdown HT29 and SW480 cell suspensions ($2\times10^6$ cells in 0.1 ml PBS) were subcutaneously inoculated into the right axillary fossa of mice (*n* = 6 per group). Tumor growth was measured every 3 d for 22 d, at which time the mice were sacrificed. The tumor volume (V) was calculated as follows: V = (length diameter) × (width diameter)$^2$ /2. The longest diameter did not exceed 2.0 cm, and the general condition of all the mice was good throughout the experiment. All animal experiments were approved by the Animal Care and Use Committee of Chongqing Medical University.

**Survival and gene expression analysis.** Survival analysis was performed using the starBase v2.0 website (https://starbase.sysu.edu.cn/starbase2/). This website provides users with an integrated database, including TCGA, and analysis tools for

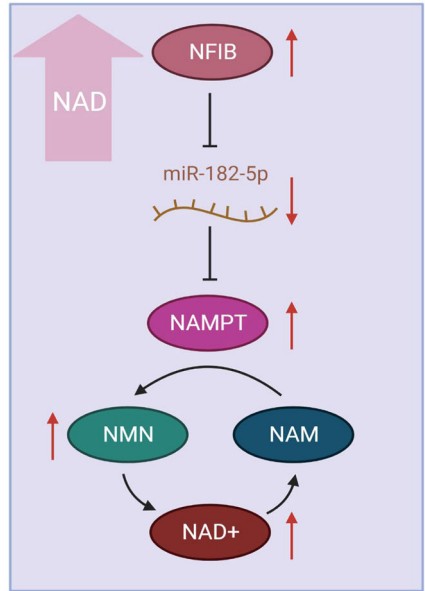
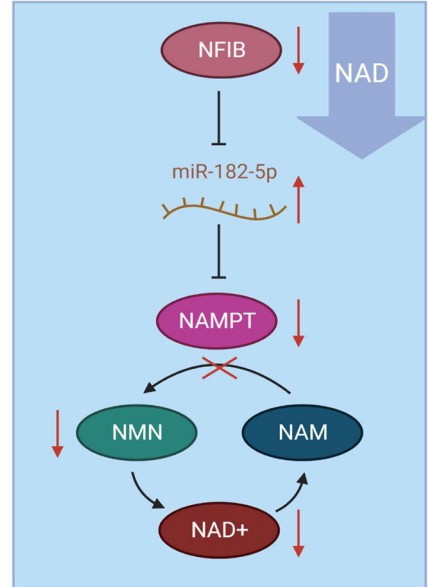

Promote CRC cells proliferation                Inhibit CRC cells proliferation

**Fig. 7 Schematic diagram of the molecular mechanism of NFIB promoting CRC cell proliferation through up-regulation of NAD+ level.** Schematic diagram of the molecular mechanism by which NFIB promotes the proliferation of CRC cells. NFIB post-transcriptionally regulates miRNA-182-5p, which then targets NAMPT to inhibit CRC cell proliferation through NAMPT-mediated of NAD+ level.

gene prognostic assessment. Statistical significance was set at p < 0.05. Gene expression analysis was performed using ENCORI and Timer 2.0 website (https://starbase.sysu.edu.cn/, http://timer.cistrome.org/). This website is mainly based on the relevant cancer data in the TCGA database for analysis and provides users with tools for gene expression analysis.

**Patients and clinical specimens**. From November 2012 to 2015, 261 pairs of CRC tissues and adjacent normal mucosa samples were collected from the Second Affiliated Hospital of Chongqing Medical University, China. All patients met the following inclusion criteria: underwent enterectomy and were confirmed by pathological diagnosis; complete clinicopathological data, including gender, age, tumor size, tumor differentiation, pT stage, pN stage, distant metastasis, pTNM, vascular invasion, and nerve invasion; complete follow-up information; and written informed consent. Patients were excluded if they received radiotherapy or chemotherapy before surgery, presented with other malignant diseases within the past 5 years, were lost to follow-up, or exhibited incomplete clinicopathological data. In addition, two cancer tissues, preferably 10 mm × 8 mm × 6 mm in size, were collected from two CRC patients. The prepared tissues were placed in a mold (Leica, 6 mm × 8 mm) and embedded in pre-cooled OCT (Sakura, Japan). Finally, the embedded tissue was transferred to a −80 °C freezer for storage and transcriptomic analysis. All study participants provided their informed consent. The study had been approved by the Second Affiliated Hospital of Chongqing Medical University and all participants signed informed consent.

**Statistics and reproducibility**. All data were analyzed using GraphPad Prism 8.0. Data are expressed as mean ± SD of three independent experiments. The significance of the in vitro and in vivo data between experimental groups was determined using the Student's t test or Mann-Whitney U test. The generalized additive model (GAM) was used to fit the curve relationship between NAMPT and NFIB in spatial transcriptome data, and spearman correlation coefficient was used to analyze the correlation. The relationship between NFIB expression and clinicopathological characteristics was assessed using the chi-squared test. Univariate Cox proportional hazard regression models were used to analyze the effects of different clinicopathological factors on overall survival. Kaplan–Meier survival curves and log-rank tests were used to determine the overall survival (OS) and disease-free survival (DFS) rates of CRC patients with different NFIB expression levels. p < 0.05 was considered to be statistically significant.

**Reporting summary**. Further information on research design is available in the Nature Portfolio Reporting Summary linked to this article.

## Data availability

Whole transcriptomic and ChIP-seq analysis data that support the findings of this study have been deposited in GEO (GSE236757). Metabolic analysis data that support the

findings of this study have been deposited in Metabolights with the identifier MTBLS7997 (www.ebi.ac.uk/metabolights/MTBLS7997). The data that support the findings of this study are available from the corresponding author upon reasonable request. Uncropped/unedited blots are provided in the Supplementary Fig. 5.

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

## Acknowledgements

This work was supported by the National Natural Science Fund (No. 81972285), Senior Medical Talents Program of Chongqing for Young and Middle-aged, Chongqing Natural Science Foundation of China (CSTB2022NSCQ-MSX0312, CSTB2022NSCQ-MSX1038), Kuanren Talents Program of the Second Affiliated Hospital of Chongqing Medical University (13-002-011, 13-004-009), Young and Middle-aged Senior Medical Talents studio of Chongqing, and Chongqing Medical Scientific Research Project (Joint Project of Chongqing Health Commission and Science and Technology Bureau, 2020FYYX103).

## Author contributions

Conception and design: Z.Z., S.C.; Development of methodology: L.Z., Z.C., H.L., S.C.; Acquisition of data: L.Z., H.L., Z.C., S.C., J.L., C.L., S.L.; Analysis and interpretation of data: L.Z., Z.Z.; Writing, review, and revision of the manuscript: L.Z., Z.Z., S.C.; Administrative, technical, or material support: Z.Z., S.H.; Study supervision: Z.Z., S.C., S.H. All authors have read and approved the final manuscript.

## Competing interests

The authors declare no competing interests.

## Additional information

**Peer review information** : *Communications Biology* thanks Gerardo Ferbeyre and the other, anonymous, reviewer(s) for their contribution to the peer review of this work. Primary Handling Editors: Mythreye Karthikeyan and Christina Karlsson Rosenthal. A peer review file is available.

