## [Peer Review File · Communications Biology]

Reviewers' comments:

Reviewer #1 (Remarks to the Author):

The authors report that NFIB promotes CRC cell proliferation by inhibiting miR-182-5p, a negative regulator of NAMPT. This interesting mechanism is supported by data showing co-expression of NAMPT and NFIB in tumor samples from colorectal cancer patients and the poorest survival for patients having both high NFIB/NAMPT.

Major issues:

- Lane 134 and fig 3f. The spatial transcriptomic data is not clear. Orange spots do not always colocalize and the data is not quantified. scRNAseq would be better to quantify the degree of co-expression between NFIB and NAMPT.
- I am not convinced that the Chip-Seq data allows to conclude that the regulation must be post-transcriptional. There are binding sites in the promoter and introns that could regulate gene expression of NAMPT.
- What are the levels of miR-182-5p in colorectal cancer either in the cohort presented in this study or in public databases.

Minor problems

- Typo: Western blot in lane 13 of supplemental data
- Figures for NAD/NADH ratios are labelled in the y axis with pmol, but what is really there is a relative amount
- In figure 2F and G, NAD/NADH ratio is used to show an increase in NAD after NMN supplementation. However, this ratio can increase because there is more NAD or less NADH, can the authors show total NAD for these experiments.
- Lanes 118-120: In the NAD biosynthesis pathway NMN bypass the need for NAMPT. However, the authors observe here that NFIB enhances conversion of NMN into NAD⁺. This is confusing after reading in the abstract that NFIB acts via NAMPT.
- Lane 122, it is premature to conclude that the proliferation effects are due to NAD synthesis at this point in the paper
- Fig 4D, y axis refers to mRNA but is miRNA
- Figure 4F: it is not consistent to only show immunoblots for HT29, please add SW480 immunoblot

Reviewer #2 (Remarks to the Author):

In this manuscript, authors identify a novel axis that may be implicated in promoting colorectal cancer (CRC) growth and proliferation, i.e. NFIB/miR-182-5p/NAMPT. Building on a prior study that demonstrated that NFIB promotes CRC tumor cell proliferation, authors embark on a well-planned journey to establish the oncogenic role of NFIB using a combination of NFIB overexpression and inhibition as well as mimics and inhibitors of miR-182-5p. First, they show that NFIB inhibition reduces a variety of cancer phenotypes, including in vivo tumor growth in a xenograft model; conversely, NFIB overexpression enhances proliferation and invasiveness. Then, they use a transcriptomic approaches to home in on important effector of NFIB, NAD and NAMPT, nicely illustrating that NFIB influences NAMPT and NAD, indirectly. Then, they identify miR-182-5p as the intermediary that directly binds to and influences NAMPT expression. Finally, using miR mimic vs inhibitors, they illustrate its control of the same cancer phenotypes as NFIB. They also add patient-relevant data on the negative prognostic implications of NFIB overexpression.

Overall, authors have done a thorough job of using both inhibitors and overexpression of key study reagents to illustrate their point. They show impact on various in vitro cancer phenotypes and incorporate 1 in vivo experiment. They also incorporate transcriptomics in isolating the

intermediary between NFIB and NMPT.

My changes are going to be minor:

Introduction

1. On page 2, authors iterate that CRC patient prognosis remains poor. Please qualify this statement by discussing reasons for CRC poor prognosis, such as metastasis at diagnosis vs. recurrence rates vs. 5-yr overall survival.
2. On page 2, authors discuss NFIB as a transcription factor of EZH2, PINK1, RIP2, ERO1A. NFIB has been shown to be regulated by c-myc in SCLC(Mollaoglu et al. *Cancer Cell* 31: 270-285 Doi 10.1016/j.ccell.2016.12.005). In CRC, myc has been shown to be associated with drug resistance and cancer stem cell phenotypes (Elbadawy et al. *Int. J. Mol. Sci.* 2019, 20(9), 2340; <https://doi.org/10.3390/ijms20092340>). Finally, NFIB silencing reduced cancer stem cell phenotype (and cancer stem cell markers) in medulloblastoma (Perumal et al. *Acta Neuropathologica Communications* 2021. <https://doi.org/10.1186/s40478-021-01299-z>). Authors could incorporate these literatures on myc-NFIB axis to strengthen their claim of the influence of NFIB on CRC aggressiveness.

Results

1. Reviewer recommends that authors start with the last result first, i.e. showing the clinical relevance of NFIB; then the in vitro and in vivo studies flow better.
2. Authors should identify the NFIB overexpressing and NFIB silenced cells lines in the body of the results.
3. How was NFIB knocked down? CRISPR vs si vs sh? Please mention in body of results.
4. Reviewer was not convinced with the H&E and Ki-67 staining presented in Supl Fig 2e.

Figures

1. Figure 6 should be Figure 1
2. Figure S2: recommend order as A, C, B, D, i.e. grouping transcriptional and translational data of the inhibition and then the overexpression.
3. What explains the high NFIB transcript level of HT29-GFP but low expression levels?

Reviewer #3 (Remarks to the Author):

In the present study, the authors described that NFIB promotes CRC growth by increasing intracellular NAD⁺ levels. NFIB post-transcriptionally upregulates the NAD⁺ salvage synthesis rate-limiting enzyme NAMPT by inhibiting miR-182-5p expression, which could target NAMPT mRNA and inhibit CRC cell proliferation. Finally, both spatial transcriptomic analysis and immunohistochemical staining demonstrated a positive correlation between NFIB and NAMPT expression in CRC specimens, and the subgroup with NFIB^{high}/NAMPT high expression had the poorest prognosis.

It is a well done study and deserves publication. Only minor scientific editings to make it clearer to readers.

Response to reviewers

We gratefully thank the editor and all reviewers for their time spend making their constructive remarks and useful suggestions, which has significantly raised the quality of the manuscript and has enable us to improve the manuscript. Each suggested revision and comment, brought forward by the reviewer was accurately incorporated and considered. Below the comments of the reviewers are replied point by point and the revisions are highlighted with yellow in documents.

Reviewer 1

1. Comment: Lane 134 and fig 3f. The spatial transcriptomic data is not clear. Orange spots do not always colocalize and the data is not quantified. scRNAseq would could be better to quantify the degree of co-expression between NFIB and NAMPT.

Reply: we appreciate your insightful suggestions, which have helped us to improve the quality of our manuscript. We further analyzed the spatial transcriptomic data, conducted co-expression analysis of NFIB and NAMPT, and obtained t-SNE and correlation analysis image. Fig. 3f and supplementary materials were adjusted (move the original fig. 3f to fig. S3b). In **fig. 3f**, we can see that in t-SNE dimension reduction, although NFIB and NAMPT spots do not completely overlap, their spatial distribution is very close with common expression (the red dots represent NFIB and the green dots represent NAMPT). Meanwhile, we statistically analyzed the correlation between NFIB and NAMPT expression in 2 CRC samples, and found that there was a positive correlation between NFIB and NAMPT ($p < 0.001$).

Since spatial transcriptomic data cannot reflect the gene expression relationship at the single-cell level, we analyzed 4 CRC tumor tissues (TNM stage: I, II, III, IV) in GSE132465 by reviewing literature and database information. We found that copy number variation (CNV) levels in the epithelial cell was higher than in the other subsets in the CNV analysis (**fig. S3d**, $p < 0.001$), indicating that the epithelial cells contained tumor cells. **Fig. 3g** showed a nonlinear positive correlation between NFIB and NAMPT in tumor cells (epithelial cells) ($R = 0.097$, $p = 1.2e-05$). We have also

modified and highlighted the body part (**lanes 132-138**).

2. Comment: I am not convinced that the Chip-Seq data allows to conclude that the regulation must be post-transcriptional. There are binding sites in the promoter and introns that could regulate gene expression of NAMPT.

Reply: We gratefully thanks for the precious time the reviewer spent making constructive remarks. First of all, we reviewed the results of ChIP-seq, as shown in fig. 3g&h, ChIP peaks were not concentrated in the transcription start site (TSS), but mainly distributed in intron and distal intergenic, it indicates that the possibility of NFIB binding with NAMPT promoter region to directly promote NAMPT transcription is low. Then, we further searched several databases (UCSC, PROMO, GeneCards and JASPAR) for transcription factors (TFs) that could bind to the NAMPT promoter region, but no NFIB was found among them, as shown in Supplementary **Table S1** for details. Finally, by reviewing the literature, we found that NAMPT can be regulated transcriptionally (TFs) or post-transcriptionally (miRNAs).

Transcription pathway: for example, **TUB** promotes NAMPT transcription in tumor-associated T cells by binding to NAMPT promoter regions (Wang Y, et al. DOI: 10.1016/j.celrep.2021.109516. Cell Rep); **MEF2C** promotes NAMPT transcription in HeLa cells by binding to NAMPT promoters (Yan SF, et al. DOI: 10.2174/13816128113199990544. Curr Pharm Des), this gene is included in our Supplementary **Table S1**; **FOXO1** directly regulates transcription of the NAMPT via binding to specific domains in the NAMPT promoter in breast cancer (Jeong B, et al. DOI: 10.1016/j.bbrc.2019.02.069. Biochem Biophys Res Commun).

Post-transcriptional pathway: such as, **miR-206** directly targets NAMPT and suppresses glycolytic activity and pancreatic cancer cell growth (Ju HQ, et al. DOI: 10.1016/j.canlet.2016.05.024. Cancer Lett); **MiR-135a** inhibits NAMPT transcription by binding to the NAMPT 3'UTR terminal, thereby inhibiting the progression of glioma (Wang J, et al. DOI: 10.1002/jcp.27946. J Cell Physiol); **MiR-26b** inhibits the expression of NAMPT via binding to its 3'-UTR in colorectal cancer cells (Zhang C, et al. DOI: 10.1371/journal.pone.0069963. PLoS One).

In conclusion, although NAMPT can be directly transcriptionally regulated by

TFs, it can also be regulated in tumors by post-transcriptional pathway, especially by miRNAs. Therefore, combined with our experimental results, we speculate that NFIB may regulate NAMPT through post-transcriptional pathway. At the same time, we have also modified and highlighted the body part (**lanes 141-147, 220-223**), and once again we would like to express our sincere thanks for your valuable suggestions.

3. Comment: What are the levels of miR-182-5p in colorectal cancer either in the cohort presented in this study or in public databases.

Reply: Thank you for your comments. By reviewing the literature and database, we found that miR-182-5p was highly expressed in corresponding adjacent normal tissues and normal intestinal epithelial cell lines compared to CRC tissues and CRC cell lines: 1. Huang W, et al. DOI: 10.3892/or.2017.5837. *Oncol Rep*; 2. Jin Y, et al. DOI: 10.26355/eurev_201902_17107. *Eur Rev Med Pharmacol Sci*; 3. Al-Sheikh YA, et al. DOI: 10.3892/ijmm.2019.4362. *Int J Mol Med*; 4. Yan S, et al. DOI: 10.1016/j.canlet.2020.04.021. *Cancer Lett*; 5. Yu DH, et al. DOI: 10.1155/2020/3981931. *Biomed Res Int*. Although two studies (4 and 18 cases) have reported that miR-182-5p is highly expressed in CRC tissues compared to tumor-adjacent tissues, they also found no significant difference in miR-182-5p expression in colorectal adenocarcinoma and low/moderate-differentiated CRC compared to tumor-adjacent tissues. Therefore, miRNA-182-5p is low expressed in CRC in most studies. A few studies suggested high expression, but the sample size was small. We added this section to the discussion part and highlighted it (**lanes 232-238**).

4. Comment: Typo: Western bolt in lane 13 of supplemental data.

Reply: Thanks for your detailed correction. I apologize for our negligence. We have corrected the errors and highlighted it in the supplementary data.

5. Comment: Figures for NAD/NADH ratios are labelled in the y axis with pmol, but what is really there is a relative amount.

Reply: Thanks for your comments, and we strongly agree with it. We quantitatively converted cell NAD⁺ and NADH according to the instructions of the NAD/NADH assay kit (abcam, ab65348). Through the standard curve, we obtained the content

units of NAD⁺ and NADH in pmol, but NAD/NADH in figures are a ratio. In order to avoid ambiguity, all pmols was deleted.

6. Comment: In figure 2f and g, NAD/NADH ratio is used to show an increase in NAD after NMN supplementation. However, this ratio can increase because there is more NAD or less NADH, can the authors show total NAD for these experiments.

Reply: We gratefully appreciate for your valuable suggestion. This is a very critical issue, which we also considered during the experiment. Take the result of HT-29 as an example:

HT29	NAD (NMN 0um)	NAD (NMN 100um)
NC	0.096	0.4265
NC	0.1195	0.473
NC	0.132	0.4945
91	0.0692	0.11408
91	0.087	0.134
91	0.0976	0.14304
93	0.048095238	0.073696145
93	0.057857143	0.089909297
93	0.062619048	0.097732426

As can be seen from the above table and figure, the NAD level of each group increased compared with the control group after the addition of NMN (100 μ m). The transformation of NAD⁺ (oxidized coenzyme I) and NADH⁺ (reduced coenzyme I) is a dynamic process. The increase of NAD⁺ content after exogenous NMN supplementation will also increase the corresponding NADH content. Therefore, the ratio of NAD/NADH can better illustrate the relationship between them and reflect the availability efficiency of NAD⁺, while showing the content of NAD⁺ alone cannot fully illustrate the strength of cell metabolic capacity. At the same time, through literature review, we also found that most of the articles use NAD/NADH to reflect the efficiency of NAD synthesis in cells (Such as, cheng Z, et al. DOI: 10.1038/nm.2049. Nat Med; Soldevila-Barreda JJ, et al. DOI: 10.1021/acs.chemrev.8b00493. Chem Rev; Diaz-Cuadros M, et al. DOI: 10.1038/s41586-022-05574-4. Nature). So we think choice the ratio of NAD/NADH is convincing.

7. Comment: Lanes 118-120: In the NAD biosynthesis pathway NMN bypass the need for NAMPT. However, the authors observe here that NFIB enhances conversion

of NMN into NAD⁺. This is confusing after reading in the abstract that NFIB acts via NAMPT.

Reply: Thanks for your careful reading. We found that the NAD⁺ content and NAMPT expression of CRC cells decreased after NFIB knockdown. According to the salvage synthesis pathway of NAD⁺ (**lanes 56-61** in the introduction), NAMPT catalyzes the conversion of NAM to NMN, which is subsequently converted from NMN to NAD⁺. Therefore, we wondered whether exogenous addition of NMN could reverse the reduction of NAD⁺ production in the case of insufficient NAMPT (hindering NMN production), thus confirming that knocking down NFIB in CRC can inhibit NAMPT expression and reduce the generation of NMN and NAD⁺. To help you understand the ideas, refer to **fig. 7** for the mechanism diagram. Thank you again for your valuable comments.

8. Comment: Lane 122, it is premature to conclude that the proliferation effects are due to NAD synthesis at this point in the paper.

Reply: Thank you for your detailed correction. We change **lane 121** "In summary, NFIB-induced NAD⁺ synthesis promoted CRC proliferation" to "In summary, NFIB promotes CRC cell proliferation by influencing the production of NAD⁺".

9. Comment: Fig 4D, y axis refers to mRNA but is miRNA.

Reply: Thank you again for your detailed correction, and I apologize for our negligence. In **fig. 4d**, "Relative mRNA of miR-182-5p" was changed to "Relative expression of miR-182-5p".

10. Comment: Figure 4F: it is not consistent to only show immunoblots for HT29, please add SW480 immunoblot.

Reply: Thanks for your comments. We have supplemented the results of SW480 in **fig. 4f**.

Reviewer 2

Introduction

1. Comment: On page 2, authors iterate that CRC patient prognosis remains poor. Please qualify this statement by discussing reasons for CRC poor prognosis, such as

metastasis at diagnosis vs. recurrence rates vs. 5-yr overall survival.

Reply: we sincerely appreciate your valuable comments. We have added the adverse factors you mentioned affecting the prognosis of CRC into manuscript to support the conclusion that the prognosis of CRC is still poor at present, and highlighted the modified part (**lanes 26-30**).

2. Comment: On page 2, authors discuss NFIB as a transcription factor of EZH2, PINK1, RIP2, ERO1A. NFIB has been shown to be regulated by c-myc in SCLC (Mollaoglu et al. Cancer Cell 31: 270-285 Doi 10.1016/j.ccell.2016.12.005). In CRC, myc has been shown to be associated with drug resistance and cancer stem cell phenotypes (Elbadawy et al. Int. J. Mol. Sci. 2019, 20(9), 2340; <https://doi.org/10.3390/ijms20092340>). Finally, NFIB silencing reduced cancer stem cell phenotype (and cancer stem cell markers) in (Perumal et al. Acta Neuropathologica Communications.2021. <https://doi.org/10.1186/s40478-021-01299-z>). Authors could incorporate these literatures on myc-NFIB axis to strengthen their claim of the influence of NFIB on CRC aggressiveness.

Reply: Thank you for your valuable advice. We have included the references you provided in the discussion to better explain the aggressiveness and mechanism of NFIB in CRC. Please see the highlights for details (**lanes 249-254**).

Results

1. Comment: Reviewer recommends that authors start with the last result first, i.e. showing the clinical relevance of NFIB; then the in vitro and in vivo studies flow better.

Reply: Thanks for your comments. As described in the abstract, our previous studies found that NFIB knockdown could affect the proliferation of CRC cells in vitro and in vivo, and then through metabonomics and NAD/NADH detection, it was found that NAD metabolism of CRC cells with NFIB knockdown was decreased. This study focuses on the effect of NFIB on the metabolism of NAD in CRC. Therefore, unlike the general flow, clinical data is placed at the end. We hope the reviewers can agree with our views, and thank you again for your valuable advice.

2. Comment: Authors should identify the NFIB overexpressing and NFIB silenced

cells lines in the body of the results.

Reply: Thank you for your careful modification. We have added the relevant information in **lanes 86 to 91** of the manuscript with highlight.

3. Comment: How was NFIB knocked down? CRISPR vs si vs sh? Please mention in body of results.

Reply: Thank you for your suggestion. We have provided a detailed description of it in the methods (**lanes 271-283**), and also included the construction method (sh) in the first part of the results (**lanes 87-91**). Thank you again for your careful reading, and I apologize for our negligence.

4. Comment: Reviewer was not convinced with the H&E and Ki-67 staining presented in Supl Fig 2e.

Reply: Thank you for your comments on our articles. The H&E and Ki-67 staining results were reviewed again. As indicated by the area circled by the red dotted line in **fig. S2e**, the xenograft tissues necrotic area in SW480-GFP group was more than that in shNFIB group (H&E). For Ki67 results (**fig. S2f**), we performed average optical density (AOD) analysis on the positive cells and drew a bar graph, which showed that the number of malignant proliferative cells in SW480-GFP group was higher than that in shNFIB group ($p < 0.01$ and 0.001). Thank you again for your patient evaluation.

Figures

1. Comment: Figure 6 should be Figure 1.

Reply: Thanks again for your comments. We have explained this in your first comment and would appreciate it if you could agree with our request to maintain the original order.

2. Comment: Figure S2: recommend order as A, C, B, D, i.e. grouping transcriptional and translational data of the inhibition and then the overexpression.

Reply: Thank you for your careful modification. We have adjusted the figures in **fig. S2** in the order of first inhibition and then overexpression.

3. Comment: What explains the high NFIB transcript level of HT29-GFP but low expression levels?

Reply: Thanks for your careful reading. In **fig. S2a&b**, HT29-GFP showed

significant differences in transcription and translation levels compared with shNFIB group. Due to the adjustment of protein loading amount and the difference in exposure time during western blot experiment, shallow bands may appear. We will pay attention to this problem in the future experiment and try our best to avoid this situation.

Reviewer 3

Thank you very much for your affirmation of our research. We have made some adjustments to the manuscript for better understanding and reading.

REVIEWERS' COMMENTS:

Reviewer #1 (Remarks to the Author):

The authors have answered very well to all reviewers' queries. In particular they provide now a statistical analysis to support their conclusions from the spatial transcriptomics data. I do recommend this paper for publication. A minor mistake should be fixed :

lane 143

but mostly located in the distal intergenic and intron regions (Fig. 4j), should be (Fig. 3j)

Reviewer #2 (Remarks to the Author):

Overall, the authors have responded to the critiques and recommendations reasonably. Minor recommendation:

Lines 87-91: Please rephrase as follows: "We then used lentiviral vectors containing shRNA sequences targeting NFIB to knock it down in CRC cell lines with high NFIB expression, i.e. HT29 and SW480 cells (Supplementary Fig. 2a, b); similarly, we overexpressed NFIB in CRC cell lines with low NFIB expression, i.e. SW620 and HCT116 cells (Supplementary Fig. 2c, d)."

Lines 249-254: Please rephrase as follows: "We found that NFIB is highly expressed in CRC, but its regulatory mechanism remains incompletely studied. C-Myc is associated with drug resistance and tumor stem cell subtypes in CRC. In SCLC, c-MYC has been reported to bind directly to the NFIB promoter and influence metastasis, while silencing NFIB reduces medulloblastoma stem cell phenotypes. Further research in CRC is needed to corroborate these findings to improve our mechanistic understanding of how NFIB influences aggressiveness in CRC"